# Aberrant Epigenetic Regulation in Head and Neck Cancer Due to Distinct EZH2 Overexpression and DNA Hypermethylation

**DOI:** 10.3390/ijms19123707

**Published:** 2018-11-22

**Authors:** Daiki Mochizuki, Yuki Misawa, Hideya Kawasaki, Atsushi Imai, Shiori Endo, Masato Mima, Satoshi Yamada, Takuya Nakagawa, Takeharu Kanazawa, Kiyoshi Misawa

**Affiliations:** 1Department of Otolaryngology/Head and Neck Surgery, Hamamatsu University School of Medicine, Hamamatsu 431-3192, Japan; daiki_m525@yahoo.co.jp (D.M.); mswyuki@abox3.so-net.ne.jp (Y.M.); imaimimi@yahoo.co.jp (A.I.); pilot1945pr@yahoo.co.jp (S.E.); tendoon@gmail.com (M.M.); veridique.star@gmail.com (S.Y.); 2Department of Regenerative and Infectious Pathology, Hamamatsu University School of Medicine, Hamamatsu 431-3192, Japan; gloria@hama-med.ac.jp; 3Department of Otorhinolaryngology/Head and Neck Surgery, Graduate School of Medicine, Chiba University, Chiba 260-8677, Japan; tnakagawa@chiba-u.jp; 4Department of Otolaryngology, Tokyo Voice Center, International University of Health and Welfare, Tokyo 107-0052, Japan; tkanazawa@iuhw.ac.jp

**Keywords:** EZH2, epigenetic regulation, DZNep, tumor-related genes, head and neck cancer

## Abstract

Enhancer of Zeste homologue 2 (EZH2) overexpression is associated with tumor proliferation, metastasis, and poor prognosis. Targeting and inhibition of EZH2 is a potentially effective therapeutic strategy for head and neck squamous cell carcinoma (HNSCC). We analyzed EZH2 mRNA expression in a well-characterized dataset of 230 (110 original and 120 validation cohorts) human head and neck cancer samples. This study aimed to investigate the effects of inhibiting EZH2, either via RNA interference or via pharmacotherapy, on HNSCC growth. EZH2 upregulation was significantly correlated with recurrence (*p* < 0.001) and the methylation index of tumor suppressor genes (*p* < 0.05). DNMT3A was significantly upregulated upon EZH2 upregulation (*p* = 0.043). Univariate analysis revealed that EZH2 upregulation was associated with poor disease-free survival (log-rank test, *p* < 0.001). In multivariate analysis, EZH2 upregulation was evaluated as a significant independent prognostic factor of disease-free survival (hazard ratio: 2.085, 95% confidence interval: 1.390–3.127; *p* < 0.001). Cells treated with RNA interference and DZNep, an EZH2 inhibitor, showed the most dramatic changes in expression, accompanied with a reduction in the growth and survival of FaDu cells. These findings suggest that EZH2 upregulation is correlated with tumor aggressiveness and adverse patient outcomes in HNSCC. Evaluation of EZH2 expression might help predict the prognosis of HNSCC patients.

## 1. Introduction

Head and neck squamous cell cancer (HNSCC) is a heterogeneous disease that potentially involves multiple sites and cellular origins within the head and neck region [1]. Despite aggressive treatments, long-term survival rates are poor and remain at ~50% [2]. Therefore, studies are required to shift their focus from prognostic biomarkers and to the development of predictive biomarkers enabling patient selection for a specific therapy [3,4]. In HNSCC, epigenomic inactivation linked to tumor suppressor genes (TSGs) are more frequent than somatic mutations in cancer and may be driving tumorigenic initiation and progression [5]. Aberrant gene promoter methylation is a key event in cancer pathogenesis and has gained increasing interest in basic and translational oncology studies because of their reversible nature [6,7]. Therefore, molecular classification of HNSCCs is required to establish prognostic and potentially preventive targets of tumor recurrence in patients.

The enhancer of zeste homolog 2 (EZH2) gene, a member of the polycomb repressor complex 2 (PRC2), contributes to the maintenance of cell identity, cell cycle regulation, and oncogenesis [8]. One of its most prominent mechanisms of action is the silencing of many cancer-related genes, thereby promoting histone H3 lysine 27 trimethylation (H3K27me3) of target gene promoters [9]. Recently, reports have shown EZH2 frequently overexpressed in several human epithelial cancers, including prostate cancer [10], breast cancer [11], gastric cancer [11], non-small-cell lung cancer [12], esophageal cancer [13], melanomas [9], oral cancer [14], and endometrial carcinomas [15]; it is additionally associated with increased tumor cell proliferation and exacerbated outcomes [16]. 3-Deazaneplanocin A (DZNep) reportedly downregulates EZH2 and inhibits H3K27me3 [17]. Although DZNep inhibits proliferation and is a highly promising antimetastatic agent in conditions such as colon cancer, lung cancer, and breast cancer [16,18,19], limited information is available regarding the effects and mechanisms of DZNep in squamous cell carcinoma [20].

Previous studies reported that EZH2 is required for the recruitment of DNA methyltransferases and facilitates CpG methylation of EZH2-target promoters [21]. Repression of the EZH2 target genes results largely from both H3K27me3 and DNA methylation, thereby providing a basis for investigating combinations of both EZH2 and DNA methyltransferase inhibitors in patients wherein EZH2 is overexpressed [22,23]. EZH2 overexpression is correlated with increased promoter methylation across tumor types in the Cancer Genome Atlas (TCGA) [24]. However, a systematic study regarding the epigenetic and transcriptional regulation of EZH2 genes in HNSCC is still needed.

This study aimed to investigate the biological effects of EZH2 regulation in HNSCC. We assessed the mRNA expression status of EZH2 target genes in HNSCC to evaluate their clinical significance as prognostic biomarkers for recurrence risk and survival. Simultaneous analyses of the EZH2 expression and the methylation of tumor suppressor genes is important to predict tumorigenesis and the development of future targeted therapy.

## 2. Results

### 2.1. EZH2 mRNA Expression Levels in Pairs of HNSCC and Normal Mucosal Tissues

EZH2 mRNA expression levels were determined in 110 HNSCC and 68 adjacent normal mucosal tissues with an independent set (original cohort). EZH2 mRNA levels were significantly greater in HNSCC tissues than in adjacent normal tissues (*p* = 0.003) (Figure 1A). Furthermore, upon comparing EZH2 expression in another cohort of 120 pairs of HNSCC and normal tissues (validation cohort), EZH2 mRNA levels in HNSCC tissues were 3-fold higher than those in paired non-cancerous mucosa (*p* = 0.002) (Figure 1B). EZH2 mRNA expression exhibited highly discriminative receiver-operator characteristic (ROC) curve profiles, which clearly distinguished HNSCC from normal mucosal tissues (area under the ROC = 0.641). At the cutoff value of 17.81, sensitivity was 33.3%; specifically, 95.1% (Figure 1C).

### 2.2. Correlation between EZH2 mRNA Expression and Clinicopathological Assessment

Characteristics and clinicopathologic features of patients are summarized in Table 1. A total of 230 tumors showed low (<17.81) and 20 high (≥17.81) EZH2 mRNA expression levels. In the original set, EZH2 mRNA levels were significantly associated with cancer stage (*p* = 0.043) and recurrence (*p* = 0.016). EZH2 expression in patients in the validation cohort was significantly associated with recurrence (*p* = 0.012). Upon combining the original and validation cohorts, EZH2 was significantly correlated with recurrence (*p* < 0.001) (Table 1).

### 2.3. Prognostic Value of EZH2 Gene Expression

Kaplan-Meier plots indicated that the expression status of EZH2 genes was related to disease-free survival (DFS) (Figure 2A–D). In the original cohort, a shorter DFS was also observed when EZH2 gene upregulation was compared with EZH2 gene downregulation (log-rank test, *p* = 0.002) (Figure 2A). EZH2 genes were upregulated in patients in the validation cohort and they had a shorter DFS than those displaying EZH2 downregulation (log-rank test, *p* = 0.024) (Figure 2B). By combining the original and validation set, DFS rates in patients displaying EZH2 upregulation was 35.5%, as compared with 56.9% in those displaying EZH2 downregulation (log-rank test, *p* = 0.0001) (Figure 2C). Moreover, among 152 patients with tumor stage T1, T2, and T3, the DFS rate in those displaying EZH2 gene upregulation was 39.8%, as compared with 55.6 % in those displaying EZH2 downregulation (log-rank test, *p* = 0.009) (Figure 2D). Furthermore, the association between methylation and risk of recurrence was estimated via multivariate analysis using a Cox proportional hazards model adjusted for age, sex, alcohol exposure, smoking status, and clinical stage. In patients displaying EZH2 upregulation (76/230, 33.0%), the adjusted odds ratio (OR) for recurrence was 2.085 (95% confidence interval [CI]: 1.390–3.127, *p* < 0.001) (Table 2).

### 2.4. Correlation between Methylation Levels of 11 Tumor Suppressor Genes and EZH2 Expression

The 11 tumor suppressor genes were defined as the number of methylated genes in each sample (Figure 3A, Appendix A). Mean differences in the methylation index of the 11 tumor suppressor genes (TS-MI) determined on the basis of EZH2 gene expression are illustrated in Figure 3B. In particular, the TS-MI was significantly higher in patients displaying EZH2 upregulation (5.97 ± 0.25) than in those with displaying EZH2 downregulation (5.21 ± 0.22, *p* = 0.032) (Figure 3B). We examined DNA methyltransferase 3α (DNMT3A) and 3β (DNMT3B) genes mRNA levels via qRT-PCR in HNSCC specimens. DNMT3A was significantly upregulated in EZH2 higher expression groups (*p* = 0.043) (Figure 3C). DNMT3B expression was not associated with EZH2 expression (*p* = 0.628) (Figure 3D).

### 2.5. siRNA-Mediated EZH2 Knockdown and DZNep Administration

We performed in vitro experiments to investigate the effect of siRNA-mediated knockdown of EZH2 protein, using the EZH2-expressing HNSCC cell line FaDu. EZH2 siRNA significantly downregulated EZH2 protein and mRNA, as evident from Western blot analysis (*p* < 0.05) (Figure 4A) and quantitative reverse transcription polymerase chain reaction (qRT-PCR) analysis (*p* < 0.05) (Figure 4B), respectively. In addition, FaDu cells with *EZH2* downregulation showed a significantly reduced colony formation ability (*p* < 0.05) (Figure 4C). As expected, DZNep treatment significantly downregulated EZH2 mRNA (*p* < 0.05) (Figure 4D). After treating cells with DZNep for 5 and 7 d, DZNep significantly inhibited cellular proliferation temporally, as evident from the growth curve (*p* < 0.05 and *p* < 0.01, respectively) (Figure 4E).

### 2.6. External Validation of Results from TCGA

Data from 43 non-tumorous tissues and 497 HNSCC tissues were analyzed. The EZH2 gene expression status was significantly higher in HNSCC than in normal samples (*p* < 0.001) (Appendix A). The Spearman analysis revealed that EZH2 expression was positively correlated with DNMT3A and DNMT3B expression (R^2^ = 0.0734, *p* < 0.001 and R^2^ = 0.02, *p* = 0.002, respectively) (Appendix A).

## 3. Discussion

This study reported that EZH2 was upregulated in HNSCC cells in comparison with normal cells, being associated with a higher recurrence risk. Based on these findings, our study is the first, to our knowledge, to report a quantifiable correlation between EZH2 overexpression and DNA hypermethylation in HNSCC. Herein, we identified that DZNep, a small-molecule EZH2 inhibitor, as a putative therapeutic target of HNSCC. These results suggest that EZH2 inhibition is a promising therapeutic avenue for a substantial fraction of HNSCC patients. Clarification of the epigenetic regulation of EZH2 gene may provide insights into the underlying mechanisms of tumorigenesis and the risk of disease recurrence in HNSCC.

EZH2 expression is reportedly associated with aggressive prostate cancer [25], esophageal cancer [13], and breast cancer [25], whereas its clinical importance in HNSCC is yet unknown. EZH2 has gained interest as a marker of aggressive subgroups in several cancers [26]. DNZep is an S-adenosylhomocysteine (SAH) hydrolase inhibitor that increases intracellular SAH levels [27]. DZNep treatment induces apotosis in chondrosarcoma cells, while decreasing their migration ability [17] [28] [29]. DZNep directly induces cellular senescence, accompanied with apoptosis, in human colon cancer cells [19]. The PRC2 inhibitory effects of DZNep are important for differentiation-inducing and anticancer activities observed in HNSCC [30].

EZH2 constitutes the catalytic subunit of PRC2 and catalyzes H3K27me3 and silences target genes via local chromatin reorganization [31]. EZH2 reportedly promotes cell proliferation, migration and invasion in different in vitro cancer cell models [32]. Viré E et al. reported that EZH2 regulates CpG methylation through direct physical contact with DNA methyltransferases [21]. EZH2 overexpression has been previously correlated with increased promoter methylation across numerous tumor types in prostate cancer, small-cell lung cancer, and clear cell sarcoma of the kidney [22,24,33]. However, a systematic study of EZH2 expression and DNA promoter methylation in most human cancers is still needed. To our knowledge, this study is the first to report EZH2 expression and DNA promoter methylation in the genesis of HNSCC.

Thus, EZH2 may play a specific role depending on the cell type and different tumorigenic sites. Such a study involving human specimens and utilizing high-throughput profiling platforms may be susceptible to measurement bias from various sources. Our results suggest, for the first time, that increased EZH2 expression is correlated with tumor progression and may facilitate the accumulation of aberrantly methylated tumor suppressor genes in HNSCC. Moreover, transcriptional activation of EZH2 was associated with aberrant methylation of other tumor-related genes and DNMT3A upregulation in HNSCC. The present results suggest that the targeted inhibition of EZH2 catalytic activity may constitute a promising therapeutic intervention; however, additional prospective studies are required to prevent broad side effects.

## 4. Materials and Methods

### 4.1. Tumor Samples

Two independent cohorts comprised 110 and 120 (original and validation cohorts) fresh frozen samples of primary HNSCC were obtained from patients from the Hamamatsu University Hospital. Written informed consent was obtained from all patients in accordance with the tenets of the Declaration of Helsinki. The study protocol was approved by the Institutional Review Board of the Hamamatsu University School of Medicine (Date of Board Approval: 2 October 2015, Code number: 25-149). Original sets of HNSCC tissues were collected from January 1995 to December 2004. For the validation set, we obtained tumor tissues and adjacent normal tissues from October 2005 to September 2016. Clinical information, including age, sex, tumor site, smoking habit, alcohol consumption, tumor size, lymph node status, and stage grouping were obtained from the patients’ clinical records. Appendix A summarizes the demographic characteristics of the study population.

### 4.2. Cell Culture

The SCC cell line FaDu was purchased from the American Type Culture Collection (ATCC, Manassas, VA, USA). Cells were cultured in Dulbecco’s modified Eagle’s medium (FUJIFILM Wako Pure Chemical Corporation, Osaka, Japan) supplemented with 10% fetal bovine serum (ThermoFisher Scientific Inc., Waltham, MA, USA) and 1% penicillin/streptomycin (FUJIFILM Wako Pure Chemical Corporation) in a humidified atmosphere containing 5% CO_2_ at 37 °C.

### 4.3. RNA Extraction and Quantitative Reverse Transcription PCR (qRT-PCR)

Total RNA was isolated using an RNeasy Plus Mini Kit (Qiagen, Hilden, Germany); cDNA was synthesized using a ReverTra Ace qPCR RT Kit (Toyobo, Tokyo, Japan). EZH2, DNMT3A, DNMT3B, and GAPDH mRNA expression levels were measured via qRT-PCR using SYBR Premix Ex Taq (Takara Bio Inc., Tokyo, Japan), the Takara Thermal Cycler Dice Real Time System TP8000 (Takara Bio Inc.), and the primer sets presented in Appendix A. Target gene expression levels were normalized to those of GAPDH using the 2^−ΔΔ^Ct method.

### 4.4. EZH2 siRNA Transfection, Colony forming Assay, and Cell Proliferation Assay

EZH2 siRNA (EZH2HSS176653, Invitrogen, Thermo Fisher Scientific, MA, USA) was diluted in Opti-MEM I Reduced Serum Medium (Invitrogen) and Lipofectamine RNAiMAX Transfection Reagent (Invitrogen). siRNA complexes were transfected into cells at a final concentration of 50 nM for RNA and protein assays, 20 nM for the colony forming assay, and 0 nM in the control cells, in accordance with the manufacturer’s instructions. The EZH2 functional inhibitor, DZNep (Sigma-Aldrich, St. Louis, MO, USA), was added to FaDu medium at 10 μM for the cell proliferation assay. For the colony forming assay, cells were seeded in 6-well plates at a density of 0.05 × 10^4^ cells per well and cultured for 14 d, followed by fixation and staining 40% ethanol-1% crystal violet and enumerated in a field of >0.1 mm diameter. For the cell proliferation assay, cells (0.2 × 10^4^ cells) were placed into each well in 12-well plates enumerated and at 3 time points (day 2, day 5, and day 7).

### 4.5. Western Blot Analysis

Cells were lysed using RIPA Buffer (FUJIFILM Wako Pure Chemical Corporation) containing Protease Inhibitor Cocktail (Promega, Madison, WI, USA). The supernatant was harvested, and protein content was measured using the Protein Assay Rapid Kit WAKO II (FUJIFILM Wako Pure Chemical Corporation). The proteins were separated via sodium dodecyl sulfate polyacrylamide gel electrophoresis (SDS-PAGE) on 7.5% TGX FastCast Acrylamide Kit gels (Bio-Rad Laboratories, Inc., Hercules, CA, USA) and electroblotted onto Immobilon-P polyvinylidene difluoride membranes (Merck KGaA, Darmstadt, Germany). The membranes were incubated overnight with anti-EZH2 antibody (Cat. No. 612667, BD Biosciences, Franklin Lakes, NJ, USA) and anti-β-actin antibody (Cat. No. A2228, Merck KGaA) at 4 °C, followed by incubation with biotin-conjugated secondary anti-mouse antibody (Nichirei Biosciences Inc., Tokyo, Japan) followed by the horseradish peroxidase-conjugated avidin-biotin reaction (Nichirei Biosciences Inc.). Immunoreactive bands were visualized using an enhanced chemiluminescence substrate (GE Healthcare UK Ltd, Buckinghamshire, UK). Western blot images were obtained using a ChemiDoc Touch imaging system (Bio-Rad Laboratories, Inc.) and analyzed using Image Lab 5.2 (Bio-Rad Laboratories, Inc.).

### 4.6. DNA Extraction, Bisulfite Treatment, and Quantitative Methylation-Specific PCR (QMSP) Analysis

Genomic DNA was extracted from tumor specimens harvested from 216 primary HSNCC patients and subjected to bisulfite conversion using the MethylEasy Xceed Rapid DNA Bisulfite Modification Kit (Takara Bio Inc.) in accordance with the manufacturer’s instructions. QMSP was performed as described previously [34]. The QMSP primer sequences for CCBE1, CDH1, COL1A2, DAPK, GALR1, MGMT, p16, RASSF1A, SALL3, SST, and TAC1 are provided in Appendix A. Overall methylation levels in individual samples were determined by calculating the methylation value. TS-MI was defined as the ratio of the number of methylated genes to the number of genes analyzed in each sample [35].

### 4.7. Collection of Publicly Available Data from TCGA

Gene expression data were obtained from the TCGA data portal (https://tcga-data.nci.nih.gov/tcga/) and MethHC (http://methhc.mbc.nctu.edu.tw/php/index.php) in July 2018 [36]. Expression data were obtained as processed RNA-seq data in the form of RNA-seq via Expectation Maximization.

### 4.8. Data Analysis and Statistics

The EZH2 expression status results and patient characteristics were compared using the students’ *t*-test. Receiver operating characteristic (ROC) curve analysis was performed using for 120 HNSCC and 120 adjacent normal mucosal samples with the Stata/SE 13.0 system (Stata Corporation, College Station, TX, USA). Area under the ROC curve analysis indicated the optimal sensitivity and specificity cut-off levels to differentiate between gene expression levels in HNSCC and normal tissue.

DFS was measured from the date of the initial treatment to the date of diagnosis of locoregional recurrence or distant metastasis. Kaplan-Meier analysis was performed to calculate survival probabilities, and the log-rank test was performed to compare survival rates. The prognostic value of EZH2 expression status was assessed via multivariate Cox proportional hazards analysis adjusted for age (≥70 versus <70 years), sex, alcohol intake, smoking status, and tumor stage (I, II, and III versus IV). Differences with *p* < 0.05 were considered significant. All statistical analyses were performed using the StatMate IV software (ATMS Co. Ltd., Tokyo, Japan).

## 5. Conclusions

In conclusion, the present results suggest that EZH2 is aberrantly expressed in HNSCC patients. CpG island hypermethylation is an early event in tumorigenesis, supporting the hypothesis that EZH2 plays an important role in tumorigenesis in HNSCC and serves as an important biomarker. This study is the first, to our knowledge, to report that EZH2 mRNA is downregulated in HNSCC owing to DNA methylation; this may be a critical event in HNSCC progression, which is associated with decreased survival.

## Figures and Tables

**Figure 1 ijms-19-03707-f001:**
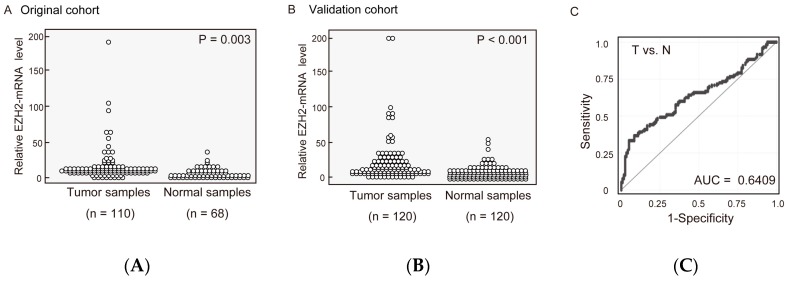
Enhancer of zeste homolog 2 (EZH2) mRNA patterns in matched pairs of head and neck squamous cell carcinoma tissue and adjacent normal mucosal tissues. Relative EZH2 mRNA expression levels in (**A**) the original cohort and (**B**) the validation cohort were analyzed in clinical samples. Changes between cancerous and normal mucosal tissues were considered significant, as determined using the Student’s *t*-test. (**C**) Area under the receiver-operator characteristic (ROC) curve (AUROC) value for EZH2 was 0.6409. At the cutoff value of 17.81, sensitivity was 33.3%; specifically, 95.1%.

**Figure 2 ijms-19-03707-f002:**
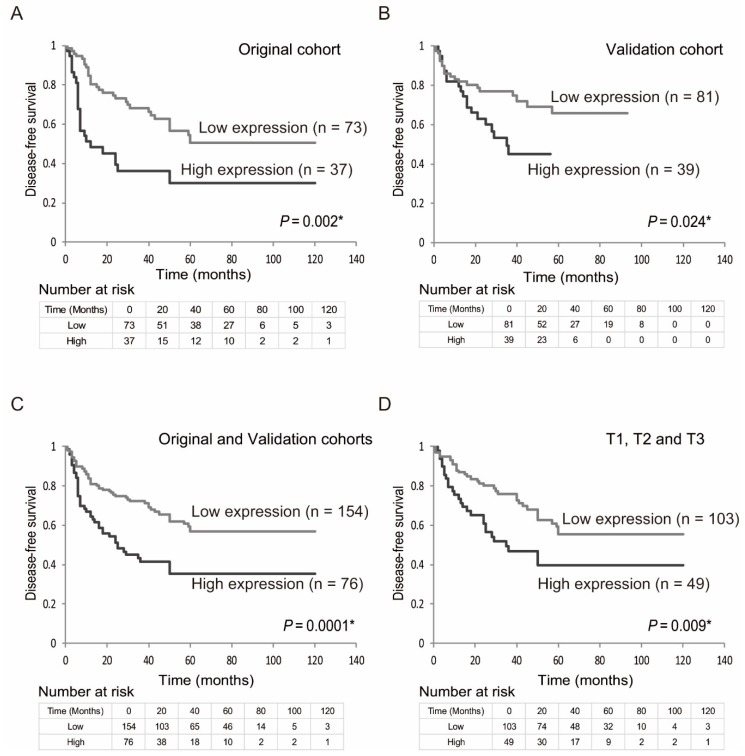
Kaplan-Meier survival curves for patients with head and neck squamous cell carcinoma (HNSCC), based on EZH2 gene expression status. Kaplan–Meier survival curves based on EZH2 gene expression status in patients with HNSCC. Disease-free survival based on (**A**) the original cohort, (**B**) the validation cohort, (**C**) the original and the validation cohort, (**D**) and the original and the validation cohort in tumor stage 1, 2, and 3 patients. Patients were dichotomized into low and high mRNA expression level groups by the optimal cut-off expression value. Gray and black lines indicate patients with tumors displaying EZH2 downregulation and upregulation, respectively. * *p* < 0.05.

**Figure 3 ijms-19-03707-f003:**
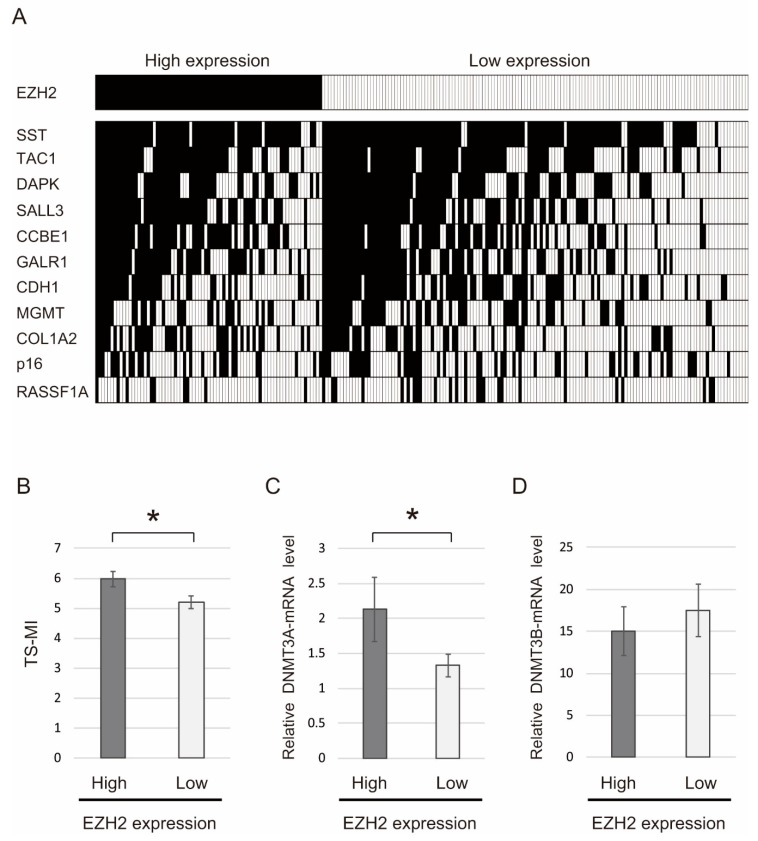
Comparison of methylation rates in 11 tumor suppressor genes along with EZH2 expression in 230 primary head and neck squamous cell carcinoma (HNSCC) tissues. (**A**) Distribution of EZH2 gene expression status and 11 tumor suppressor genes promoter methylation. Shaded boxes indicate the presence of methylation; open boxes, the absence of methylation. (**B**) Correlation between the methylation index of tumor suppressor genes (TS-MI) and EZH2 gene expression status in HNSCC patients (*p* = 0.032). The mean tumor suppressor genes (TS-MI) for the different groups were compared using a student’s *t*-test. (**C**) DNMT3A mRNA was significantly upregulated in groups displaying EZH2 upregulation (*p* = 0.032). (**D**) DNMT3B expression was not associated with EZH2 expression status (*p* = 0.628). * *p* < 0.05.

**Figure 4 ijms-19-03707-f004:**
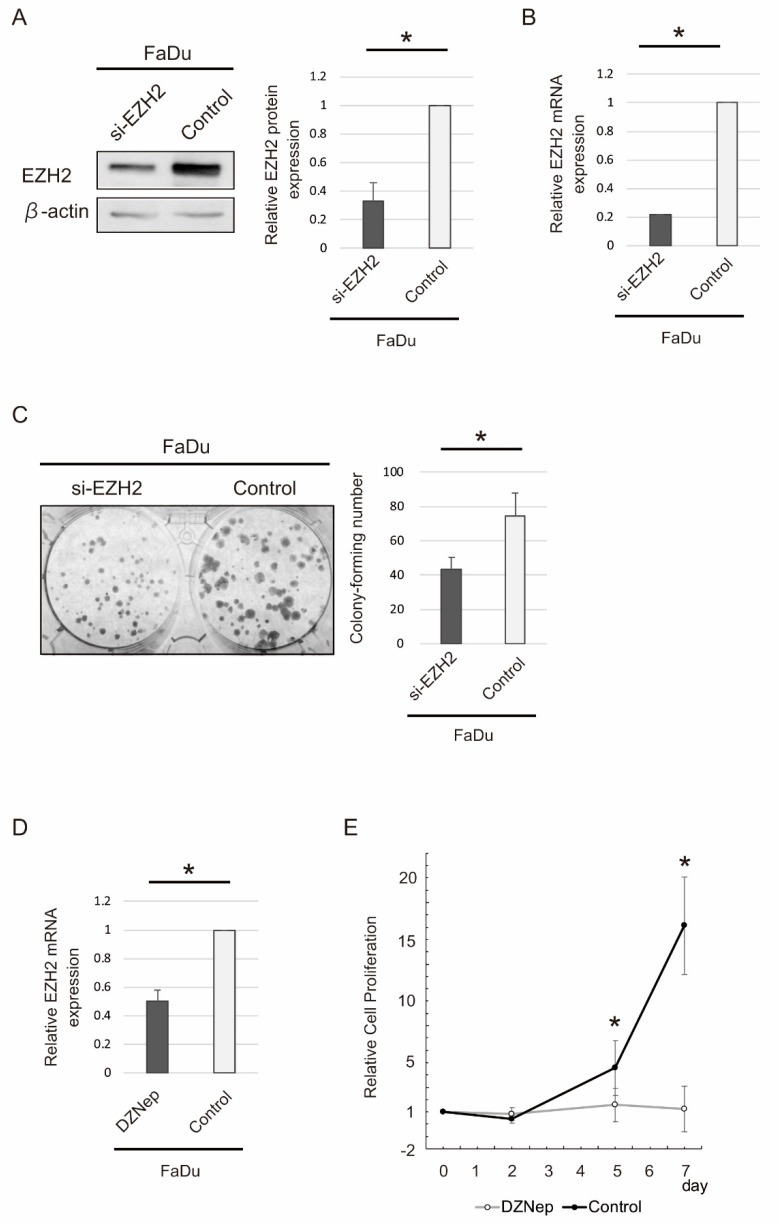
siRNA- and DZNep-mediated EZH2 knockdown inhibited the proliferation of FaDu cells. (**A**) Western blot analysis of EZH2 protein expression after siRNA transfection. (**B**) The chart illustrates the quantification of EZH2 mRNA expression levels via quantitative reverse transcription polymerase chain reaction (qRT-PCR) after siRNA-mediated knockdown of EZH2 mRNA. (**C**) Colony formation upon transfection with EZH2-targeting siRNA or control. (**D**) EZH2 mRNA expression was analyzed via qRT-PCR from FaDu cells treated with DZNep for 48 h. (**E**) Effect of DZNep (10 μM) on the growth curve. Experiments were performed in duplicate and repeated thrice. The mean ± standard deviation value for each treatment is graphed here. * *p* < 0.05.

**Table 1 ijms-19-03707-t001:** EZH2 gene expression status in (head and neck squamous cell cancer) HNSCC primary samples.

Patient and Tumor Characteristics	Original Cohort (*n* = 110)			Validation Cohort (*n* = 120)			Original and Validation Cohorts (*n* = 230)		
	High (37)	Low (73)	*p*-Value §	High (39)	Low (81)	*p*-Value §	High (76)	Low (154)	*p*-Value §
**Age**									
Under 70	25	57		26	49		51	106	
70 and Older	12	16	1	13	32	0.552	25	48	1
**Gender**									
Male	31	56		33	70		64	126	
Female	6	17	0.463	6	11	1	12	28	0.715
**Alcohol Exposure**									
Ever	21	39		29	70		50	109	
Never	16	34	0.84	10	11	1	26	45	1
**Smoking Status**									
Smoker	27	45		28	65		55	110	
Non-Smoker	10	28	0.291	11	16	1	21	44	1
**Tumor Size**									
T1–3	22	56		27	48		49	104	
T4	15	17	1	12	33	0.321	27	50	1
**Lympho-Node Status**									
N0	15	36		18	37		33	73	
N+	22	37	0.423	21	44	1	43	81	0.578
**Stage**									
I, II, III	12	40		20	32		32	72	
IV	25	33	0.043 *	19	49	0.243	44	82	0.574
**Recurrence Events**									
Positive	25	31		19	21		44	52	
Negative	12	42	0.016 *	20	60	0.012 *	32	102	<0.001 *

§ Fisher’s exact probability test; * *p* < 0.05.

**Table 2 ijms-19-03707-t002:** Multivariate analysis of factors affecting survival using Cox proportional hazards model in 230 HNSCC patients.

Variables	Disease-Free Survival	
	HR (95% CI)	*p*
**Age**		
70 and older vs. <70	1.326 (0.860–2.044)	0.202
**Gender**		
Male vs. Female	1.065 (0.615–1.846)	0.821
**Alcohol exposure**		
Ever vs. Never	0.516 (0.308–0.865)	0.012 *
**Smoking status**		
Smoker vs. Non Smoker	1.988 (1.127–3.505)	0.018 *
**Stage**		
I, II, III vs. IV	1.672 (1.088–2.568)	0.019 *
**EZH2 Expression**		
High vs. Low	2.085 (1.390–3.127)	<0.001 *

HR: hazard ratio; 95% CI: 95% confidence interval; * *p* < 0.05.

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
