# Peer review of "Aberrant Epigenetic Regulation in Head and Neck Cancer Due to Distinct EZH2 Overexpression and DNA Hypermethylation"

_ijms, 2018, doi:10.3390/ijms19123707_

Reviewer 1 Report

Mochizuki et al present an interesting study on the potential role of EZH2 in HSNCC, focusing on both primary tumor analysis and some cell line manipulation.

-Figure 1: Could we see the data presented in a dot plot rather than bar graph? I.e. plot each expression value as a dot and compare tumors to normals? Also, what is this normalized to? Reference gene expression?
-Table 1: Needs legend. How did they decide “High” versus “Low” expression? Unclear from this table. Same applies to Figure 2.
-Figure 3: How did they pick these 11 TS genes for methylation analysis? Are they implying that EZH2 affects DNMT levels? If so, they should assess both mRNA and protein levels of all three DNMTs (DNMT1, DNMT3A, DNMT3B) and see if expression levels of these enzymes affects tumor suppressor methylation independently of EZH2 expression.
-Figure 4: Results are promising but the study would be more convincing if they used more than one cell line and got similar results.

Author Response

Dear Editor:
We appreciate the opportunity to revise our manuscript entitled “Aberrant epigenetic regulation in head and neck cancer due to distinct EZH2 overexpression and DNA hypermethylation.” (Manuscript ID, ijms-385501). We have included our point-by-point responses to the reviewers’ comments and questions below. All changes are highlighted in red in the revised manuscript. We hope that our manuscript is now acceptable for publication in the International Journal of Molecular Sciences.

Re: Manuscript ID: ijms-385501

Thank you for your letter dated November 2nd that gave us the opportunity to revise our manuscript in response to the reviewers’ comments. We have provided a point-by-point responses to the reviewers’ comments and questions below.

Reviewer #1 (Comments to the Author):

Mochizuki et al presents an interesting study on the potential role of EZH2 in HSNCC, focusing on both primary tumor analysis and some cell line manipulation.

Our Response: We thank the reviewer for the careful review of our manuscript. Please see our responses to the comments below and the corresponding changes in our manuscript (in red).

Figure 1: Could we see the data presented in a dot plot rather than bar graph? I.e. plot each expression value as a dot and compare tumors to normals? Also, what is this normalized to? Reference gene expression?

Our Response: We changed to plotted display of QPCR analysis for the patient samples, as suggested by the reviewer. These data have been presented in the new Figure 1. We are using GAPDH as control housekeeping gene, to normalize the value of target genes.

Table 1: Needs legend. How did they decide “High” versus “Low” expression? Unclear from this table. Same applies to Figure 2.

Our Response: Thank you for your comments. This sentence has been added to the Figure 2 legend beginning on page 5, line 123, as follows: Patients were dichotomized into low and high mRNA expression level groups by the optimal cut-off expression value. On page 3, line 95, we have added the following results explains: A total of 230 tumors showed low (<17.81) and 20 high (≥17.81) EZH2 mRNA expression levels.

Figure 3: How did they pick these 11 TS genes for methylation analysis? Are they implying that EZH2 affects DNMT levels? If so, they should assess both mRNA and protein levels of all three DNMTs (DNMT1, DNMT3A, DNMT3B) and see if expression levels of these enzymes affects tumor suppressor methylation independently of EZH2 expression.

Our Response: The reviewer has raised a good issue. Our research groups mainly research about epigenetic change of neuropeptide genes and G protein-coupled receptor (GPCR) genes. In addition, we examined methylation levels of some TS genes. This study focused these 11 TS genes that were of great prognostic significance in HNSCC. Although this study did not include EZH2 and DNMTs protein level analysis of HNSCC patients, some evidence has been suggested that the EZH2 may act as a potent epigenetic regulator. Hence, mRNA expression analysis by using real time PCR is best suited for clinical studies. Receiver operating curve analysis has become a popular method for evaluating the accuracy of medical diagnostic systems. By measuring various parameters, such as the area under the curve, it assesses the inherent ability of the system to discriminate between diseased and healthy populations or samples thereof and identifies optimal cutoff points. We regret our inability to accomplish this request for inclusion in the current manuscript.

Figure 4: Results are promising but the study would be more convincing if they used more than one cell line and got similar results.

Our Response: The reviewer has raised an important question. Other HNSCC cell line in our laboratory that was the EZH2 low expressing, was not suitable for experiments to investigate the effect of siRNA-mediated knockdown of EZH2 protein. We nonetheless believe that our data are worth reporting because they can be considered as hypothesis-generating and may stimulate further research in the field.

Reviewer 2 Report

Daiki Mochizuki in the manuscript titled “Aberrant epigenetic regulation in head and neck (HNSCC) reports cancer due to distinct EZH2 overexpression and DNA hypermethylation” reports the expression analyses of EZH2 gene and methylation directly in HNSCC patient tissues. They further reproduced the work done by other researchers on the effect of EZH2 inhibitors on head and neck squamous cell carcinoma

Minor:

Authors have not cited original papers and instead used reviews for citations purposes. This reviewer would like to see the original citations.
Authors also missed important citations such as:
Stransky N, Egloff AM, Tward AD, et al. The mutational landscape of head and neck squamous cell carcinoma. Science 2011;333:1157–60.
Kidani K, et al.High expression of EZH2 is associated with tumor proliferation and prognosis in human oral squamous cell carcinomas. Oral Oncol 2009;45:39–46
Tan J, et al. Pharmacologic disruption of polycomb-repressive complex 2-mediated gene repression selectively induces apoptosis in cancer cells. Gene Dev 2007;21:1050–63.
Oncotarget. 2015 Oct 20;6(32):33720-32. doi: 10.18632/oncotarget.5606.

Major:

TCGA datasets should be analyzed for EZH2 expression, DNA methylation of reported genes and epigenetic modifiers. This will increase the overall significance of the study.

Author Response

Dear Editor:
We appreciate the opportunity to revise our manuscript entitled “Aberrant epigenetic regulation in head and neck cancer due to distinct EZH2 overexpression and DNA hypermethylation.” (Manuscript ID, ijms-385501). We have included our point-by-point responses to the reviewers’ comments and questions below. All changes are highlighted in red in the revised manuscript. We hope that our manuscript is now acceptable for publication in the International Journal of Molecular Sciences.

Re: Manuscript ID: ijms-385501

Thank you for your letter dated November 2nd that gave us the opportunity to revise our manuscript in response to the reviewers’ comments. We have provided point-by-point responses to the reviewers’ comments and questions below.

Reviewer #2 (Comments to the Author):

Daiki Mochizuki in the manuscript titled “Aberrant epigenetic regulation in head and neck (HNSCC) reports cancer due to distinct EZH2 overexpression and DNA hypermethylation” reports the expression analyses of EZH2 gene and methylation directly in HNSCC patient tissues. They further reproduced the work done by other researchers on the effect of EZH2 inhibitors on head and neck squamous cell carcinoma

Our Response: We thank the reviewer for the positive review of our paper.

Minor:
Authors have not cited original papers and instead used reviews for citations purposes. This reviewer would like to see the original citations.
Authors also missed important citations such as:
Stransky N, Egloff AM, Tward AD, et al. The mutational landscape of head and neck squamous cell carcinoma. Science 2011;333:1157–60.
Kidani K, et al.High expression of EZH2 is associated with tumor proliferation and prognosis in human oral squamous cell carcinomas. Oral Oncol 2009;45:39–46
Tan J, et al. Pharmacologic disruption of polycomb-repressive complex 2-mediated gene repression selectively induces apoptosis in cancer cells. Gene Dev 2007;21:1050–63.
Oncotarget. 2015 Oct 20;6(32):33720-32. doi: 10.18632/oncotarget.5606.

Our Response: Thank you for this suggestion. I have read the papers of Stransky N et al. (Science 2011), Kidani K et al. (Oral Oncol 2009), Tan J et al. (Gene Dev 2007), and Zhou X et al. (Oncotarget. 2015). We have added a description of some of their results and information to the introduction and discussion as a description of how EZH2 results in HNSCC tumorigenesis.

Major:
TCGA datasets should be analyzed for EZH2 expression, DNA methylation of reported genes and epigenetic modifiers. This will increase the overall significance of the study.

Our Response: The reviewer has raised a good point. We performed the validation analysis suggested by the reviewer. We have added these data as a new Figure S2.

Round  2

Reviewer 1 Report
The authors have addressed my questions and concerns and I find the manuscript suitable for publication.